# Trends and Changes in Hydrologic Cycle in the Huanghuaihai River Basin from 1956 to 2018

Xiaotian Yang [1,2], Zhenxin Bao [2,3,*], Guoqing Wang [2,3], Cuishan Liu [2,3] and Junliang Jin [2,3]

1   College of Hydrology and Water Resources, Hohai University, Nanjing 210098, China; 191301010059@hhu.edu.cn
2   Research Center for Climate Change, Ministry of Water Resources, Nanjing 210029, China; gqwang@nhri.cn (G.W.); csliu@nhri.cn (C.L.); jljin@nhri.cn (J.J.)
3   State Key Laboratory of Hydrology-Water Resources and Hydraulic Engineering, Nanjing Hydraulic Research Institute, Nanjing 210029, China
*   Correspondence: zxbao@nhri.cn

**Abstract:** The Huanghuaihai River Basin (HRB) is one of the most prominent areas of water resource contradiction in China. It is of great significance to explore the relationship between water balance in this area for a deep understanding of the response of the water cycle to climate change. In this study, machine learning methods are used to prolong the actual evapotranspiration (ET) of the basin on the time scale and explore water balances calculated from various sources. The following conclusions are obtained: (1) it is found that the simulation accuracy of Global Land Evaporation Amsterdam Model (GLEAM) products in HRB is good. The annual average ET spatial distribution tends to increase from northwest to southeast; (2) three machine learning algorithms are used to construct the ET calculation model. The correlation coefficients of the three methods are all above 0.9 and the mean relative error values of random forest (RF) are all less than 30%. The RF has the best effect; (3) the relative errors of water balance in HRB from 1956–1979, 1980–2002 and 2003–2018 are less than ±5%, which indicates that the calculation of each element of the water cycle in the study area can well reflect the water balance relationship of the basin.

**Keywords:** actual evapotranspiration; machine learning; hydrologic cycle; Huanghuaihai River Basin

## 1. Introduction

Water balance refers to the relationship between water budget and storage formed in the water cycle process on a certain space-time scale. The basic expression is the water balance equation [1]. The constituent elements include replenishment, excretion, consumption and storage variation, that is, source sink and state variables. The balanced relationship of regional water plays an important role in indicating the carrying state of water resources, and the abnormal changes, which exceed the threshold, will lead to ecological environment and socio-economic problems [2]. In the normal equilibrium state, the storage variables fluctuate up and down, but close to zero on a multi-year scale; in the non-equilibrium state, the storage variables show irreversible changes. The water balance equation expresses the total water input in the form of precipitation in the watershed as the amount of water returned to the atmosphere by evapotranspiration (ET), the amount of water flowing out of the catchment in the form of runoff, and the change in terrestrial water storage [3]. The water balance equation is a calculation equation that follows mass conservation. It is an indispensable tool to verify the water cycle in the basin scale.

Some studies have used the water balance equation to explain hydrological climate change in the basin [4] and verify the accuracy of the estimation of one component, as well as the estimation of some components when other components are known [5,6]. In the study of the water balance relationship, the accuracy of the results will be limited by the uncertainty of each variable in the water balance equation. Especially in large spatial

scales and less-developed areas, there will be water imbalance due to the lack of data from measured stations, as well as uncertainties caused by measuring instruments and methods [7]. For each component of the water balance, such as precipitation, ET, runoff, and water storage, flux data assimilation techniques can be optimized to combine information from multiple data sources (field observations, satellite remote sensing, surface models, and reanalysis data). They can also spatially optimize the combination of multiple available data sources for changes in precipitation, ET, runoff, and total water storage [8]. Globally, water balance is achieved through data assimilation techniques [9,10]. By comparing the calibrated data set with the ground-based observation results, the combined data sets of precipitation, ET and runoff are usually superior to a single data set [11].

In previous studies, precipitation and runoff have been observed for a long time in many watersheds, and there are many studies on them [12–14]. However, because of the limitation of observation data, there are few studies on the variation in ET and terrestrial water storage (TWS), especially for a long time. ET mainly includes soil evaporation, vegetation transpiration and water surface evaporation. It is the key element of ground-gas exchange, the thermodynamic process and atmospheric dynamic process, and its uncertainties are also the most important among the components of land water cycle [15]. Affected by climate, vegetation and many kinds of environmental factors [16], how to regard ET as an important link in the water cycle and realize accurate estimation of regional ET has always been a difficult point in the field of hydrometeorology [17,18]. Hydrological model simulation is a common method to calculate each component in water cycle [19]. However, the input and output data, model structure, initial conditions and model parameters of the hydrological model will cause the uncertainties of the model and affect the accuracy of the simulation. In order to improve the accuracy and spatial resolution, the remote sensing technology developed in recent years can provide the instantaneous value of terrestrial parameters, and provide the spatial continuous estimation of each component of the land water cycle from region to global [20]. However, whether the estimates of these individual components are accurate enough or not [21,22] and how to estimate variations in ET and TWS before the availability of remote sensing data still need to be studied.

As one of the most contradictory areas of water resources in China, the Huanghuaihai River Basin (HRB) has complex surface conditions, including various types of ecosystem, multi-climatic zones and so on. There are great uncertainties about the value and trend of ET [23]. Improving ET estimation in HRB is very important. Combining various ET algorithms with machine learning methods, we propose a method of extending region ET estimation. We collected the data of precipitation, ET, latent heat flux, surface water resources and groundwater resources of HRB from 1956 to 2018. According to the data of flux station, we evaluated the accuracy of the terrestrial evapotranspiration dataset across China (TEDAC), Global Land Evaporation Amsterdam Model (GLEAM) and Global Land Data Assimilation System (GLDAS) data sets. The study applied machine learning methods to prolong ET of HRB on the time scale. Meanwhile, combined with the change in water reserves, we analyzed the error of water balance calculations according to different data sets from the view of water balance, then the evolution of each element of the water cycle and the change in water balance are analyzed systematically. It is crucial to improve the carrying capacity of regional water resources and realize the rational allocation of water resources.

## 2. Study Area and Data

### 2.1. Study Area

The HRB consists of three major water resources areas, the Haihe River Basin, Huaihe River Basin and Yellow River Basin. The study area lies between 96°–123° E and 32°–43° N (Figure 1). The total area of the study area is about 1.433 million km$^2$, accounting for 14.8% of the territory of China. There is a shortage of water resources in the HRB, which accounts for 7% of the country's water resources and carries 34% of the country's population and 38% of GDP. Because of the vast area and complex terrain, the climate conditions in this

area are complex and changeable, and the monsoon climate is the main feature of the area. According to the topographic and geomorphological characteristics of HRB, 59 tertiary areas in the basin are divided into 8 regions, namely, Haihe mountainous area, Haihe plain, Huaihe River Basin, Shandong Peninsula, area above Lanzhou, Lanzhou-Tudaoguai Interval, Middle Yellow River and Lower Yellow River. In recent years, due to the abnormal change in global climate, the water circulation system has changed to a certain extent, which affects the relationship between the water resources situation and water balance. Therefore, it is necessary to study the relationship between the evolution of water cycle elements and water balance in HRB. This is of great significance for analyzing the impact of climate change on ET and TWS, improving the understanding of the impact of climate change, and exploring the development and utilization of water resources in the basin.

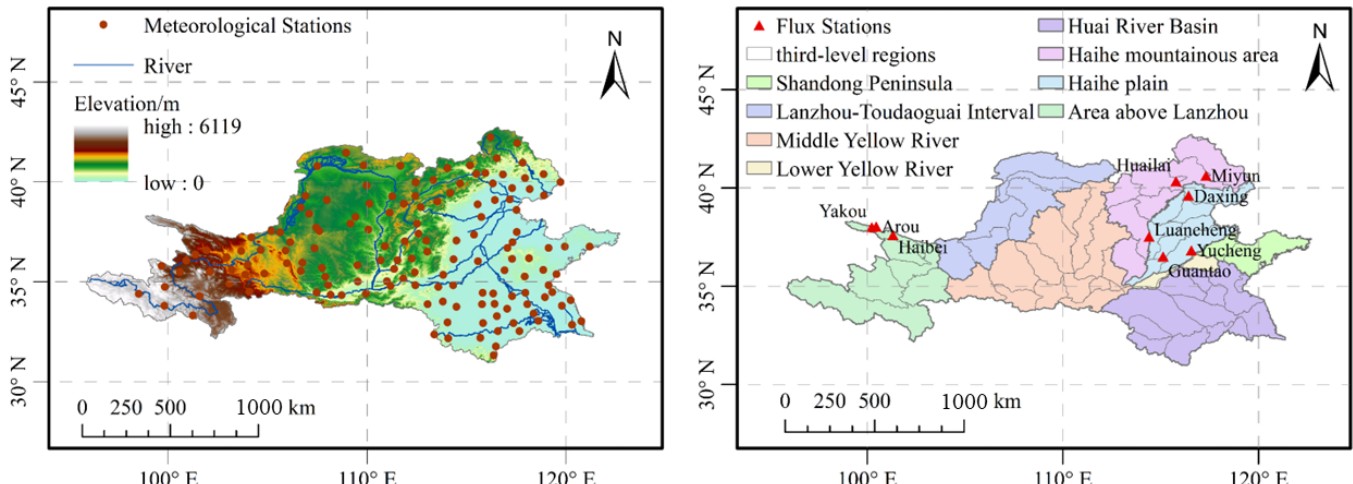

**Figure 1.** Zoning of Huanghuaihai River basin and distribution of meteorological stations.

*2.2. Data Sources*

2.2.1. Hydrological and Meteorological Data

The meteorological data used in this paper are derived from the Daily Data Set of China Surface Climate data (V3.0), which includes the daily observation data of precipitation, temperature, evaporation, sunshine, wind direction and wind speed, station pressure, relative humidity and 0 cm ground temperature from 699 meteorological stations in China from 1951 to 2018. The data set has been strictly controlled by quality. It is widely used in the field of hydrometeorology because of its high integrity, continuity and reliability. The data acquisition site is as follows: http://data.cma.cn/, accessed on 27 September 2021. The location of the weather station is shown in Figure 1. The spatial scale of this study is 59 tertiary basins in HRB, while the spatial distribution of more than 180 meteorological stations in HRB is very uneven.

In this study, the meteorological data of meteorological stations in the basin are interpolated by the inverse distance weight interpolation method. The interpolation data of each grid center point are approximately replaced by the surface data of each grid according to the plane relationship between the grid and the HRB. According to the plane relationship between the grid and the HRB, the meteorological data of each three levels of watershed are obtained by using the area weighted average. The interpolation of meteorological data from point to surface is realized.

2.2.2. Evapotranspiration Product Data

GLEAM [24] is a set of algorithms based on remote sensing observation data to estimate different surface evaporation components (including transpiration, bare soil evaporation, interception loss, snow sublimation and open water evaporation). The intermediate output of the model algorithm also includes potential evaporation, soil moisture in the vegetation root area and surface soil moisture. The basic principle of the algorithm is to

make maximum use of a climate remote sensing observation data set to reverse evaporative information. Firstly, the potential evaporative emission is calculated according to the observation data of net radiation and near-surface temperature and Priestley–Taylor equation. The evaporation limiting factor is calculated by considering vegetation water content and root soil moisture, so that the potential evaporative emission is converted into actual evaporative emission. The actual evapotranspiration data of GLEAM used in this study are version v3.3a. The spatial resolution is $0.25° \times 0.25°$ and the time span is 1980–2018.

TEDAC [25] is a Chinese surface evaporative product (v1.5) based on the evaporative complementary method. The input data include downward shortwave radiation, downward long wave radiation, air temperature, air pressure of China Meteorological Forcing Dataset (CMFD), surface emissivity and albedo of Global Land Surface Satellite (GLASS), surface temperature and air humidity of ERA5-land, scattering emissivity of National Centers for Environmental Prediction (NCEP) and so on. The time span of the dataset is from 1982 to 2017 and the spatial resolution is 0.1°. The temporal resolution is monthly and the spatial range is the land area of China. The dataset provides a basis for the study of long-time scale water cycles and climate change.

GLDAS is a kind of land surface hydrological model that is widely used at present. GLDAS-2.0 and GLDAS-2.1 data are provided by using surface observation data and satellite remote sensing data as driving data, combined with four land surface models of CLM, Noah, Mosaic and VIC. In this paper, the monthly evaporative data of GLDAS-2.1 NOAH dataset are used. The time span is 2000–2018 and the spatial resolution is $0.25° \times 0.25°$.

The flux tower data (Table 1) adopt the latent heat flux observation data released by the eddy correlator. This data set is provided by the "National Cryosphere Desert Data Center/National Service Center for Speciality Environmental Observation Stations". The average period of the observed data was 30 min, 48 data a day, and the missing data were marked as −6999. Data processing includes the following:

(i) Data quality control. The quality control steps of data include eliminating outliers, correcting delay time, coordinate rotation (plane fitting method) and frequency response correction. Ultrasonic virtual temperature correction and density correction include eliminating the data of instrument error; eliminating the data of 1 h before and after precipitation; eliminating the data of 10 Hz original data with a loss rate of more than 3% per 30 min; and eliminating the observed data of weak turbulence at night.

(ii) Data interpolation. When the missing value is less than 2 h, the effective flux before and after the time is calculated by linear interpolation. When the missing value is between 2 h and 4 d, the average diurnal variation method is used to calculate the average effective flux in the same period of 5 days before and after the adjacent period. When the missing measurement value exceeds 4 days, it may be caused by power supply or instrument failure. The linear fitting relationship between latent heat flux and air temperature, sensible heat flux and net radiation is established by selecting the observation data of the vacancy adjacent period, and the daily value is inserted according to the simulated value.

(iii) Calculation of evaporation. This is carried out according to latent heat flux and vaporization latent heat of water ($\lambda$ = 2.45 MJ $\times$ kg$^{-1}$). The actual evaporation of the basin is calculated by Formula (1).

$$ET = \frac{LE}{\lambda} \tag{1}$$

The site distribution is as follows.

**Table 1.** Basic information of flux stations.

| Type | Site | Longitude | Latitude | Underlying Surface Type | Altitude/m | Time Series |
|---|---|---|---|---|---|---|
| Grassland | Haibei | 101.3 | 37.6 | Alpine meadow | 3216 | 2003–2010 |
| | Arou | 100.5 | 38.0 | Alpine meadow | 3033 | 2015–2017 |
| | Yakou | 100.2 | 38.0 | Alpine meadow | 4148 | 2015–2017 |
| Farmland | Luancheng | 114.4 | 37.5 | Double cropping rotation of Winter Wheat and summer maize in one year | 50 | 2007.10–2013.9 |
| | Daxing | 116.4 | 39.6 | Corn/wheat, melon and fruit | 20 | 2008–2010 |
| | Guantao | 115.1 | 36.5 | Corn/wheat, cotton | 30 | 2008–2010 |
| | Miyun | 117.3 | 40.6 | Orchards, corn/bare land, towns | 350 | 2008–2010 |
| | Huailai | 115.8 | 40.4 | Watered field corn | 480 | 2013–2018 |
| | Yucheng | 116.6 | 36.8 | Warm temperate semi-humid dry farmland | 28 | 2003–2010 |

### 2.2.3. Terrestrial Water Storage Data

At present, Gravity Recovery and Climate Experiment (GRACE) data are mainly calculated and published by the Center for Space Research at the University of Texas (CSR), Jet Propulsion Laboratory (JPL) and Geo-Forschungs-Zentrum-Potsdam (GFZ). There are two common methods to retrieve land water reserves from GRACE data. The first method is to use the spherical harmonic coefficient method to calculate water reserves and the second is to use mass concentration (Mascon) inversion. Compared with the spherical harmonic coefficient method, the Mascon product can improve the temporal and spatial resolution of the inversion results. The Mascon method is mainly based on GRACE Mascon products issued by JPL and CSR and other organizations. These data have been corrected and processed by relevant errors, which can be used to directly calculate the water reserves in the study area. In this paper, the Mascon model published by CSR is used to solve the data CSR_GRACE_GRACE-FO_RL06_Mascons, which is stored in NetCDF format with a spatial resolution of $0.25° \times 0.25°$. Based on the monthly mean value from 2004 to 2009, the negative value indicates that the water reserves in that month are lower than the monthly mean value from 2004 to 2009, and the positive value indicates that the water reserves are surplus, that is, the monthly water reserves in that month are higher than the monthly mean value from 2004 to 2009.

For the early period without GRACE data, the sum of snow water equivalent, soil moisture and plant cap water data simulated by GLDAS-2.0Noah model from January 1956 to December 2011 is used as the result of TWS. The soil moisture is divided into four layers, which are the results of soil aquifers of 0~10 cm, 10~40 cm, 40~100 cm and 100~200 cm depths. The temporal resolution is 1 month and spatial resolution is $0.25° \times 0.25°$.

### 2.2.4. Natural Runoff Data

The amount of surface water resources refers to the amount of water that can be updated year by year in all kinds of surface water bodies, including lake water, river water and ice and snow water, and can also be expressed by natural river runoff. The amount of groundwater resources is the dynamic water quantity received by underground aquifers from precipitation, surface water and so on. There is a close relationship between surface water and groundwater. The repeated parts of surface water and groundwater need to be deducted when calculating the total runoff of the basin. In this paper, the underground runoff is represented by the amount of underground non-repetition. The total runoff refers to the sum of surface and underground runoff minus the amount of repetition, excluding the amount of incoming water outside the area. The runoff data for 1956–2000 were obtained from the second national water resources assessment, and the data for 2001–2018 came from the water resources bulletins of the Haihe River, the Yellow River and the Huaihe River.

### 2.2.5. Vegetation Data

The normalized difference vegetation index (NDVI) [26] data from 1982 to 2018 are based on the NOAA meteorological satellite data set provided by NASA of the United States. The temporal resolution is 15 d and the spatial resolution is 8 × 8 km. The data set has wide coverage, long time span and strong vegetation monitoring ability. It is the longest continuous data set at present. The monthly value of NDVI is calculated by the maximum value composite (MVC) method, which effectively reduces the influence of clouds, atmosphere, aerosol and solar altitude angle in the atmosphere, and obtains quarterly or annual NDVI data according to the average value of the month.

## 3. Methodology

### 3.1. Trend Test Method

In this paper, the linear regression method is used to analyze and calculate the changing trend of time series. The regression slope can be used to characterize the variation range of time series. In addition, the Mann–Kendall (MK) test is used to calculate the statistically significant level of the comparative trend, which is widely used in meteorological and hydrological fields. When the significant level $\alpha = 0.05$ is set, the critical value of the test statistic MK value is $\pm 1.96$, that is, if MK is greater than 1.96, the increasing trend is significant at the 0.05 level. The decreasing trend is significant when the MK value is less than $-1.96$. For the detailed calculation formula and application of the MK test method, please refer to the reference [27].

### 3.2. Data Accuracy Evaluation

For the evaluation of evaporative data and machine learning algorithm, the mean absolute error (MAE), mean relative error (MRE), root mean square error (RMSE) and Pearson correlation coefficient ®are used as the evaluation basis. The formula is as follows:

$$r_{xy} = \frac{\sum_{i=1}^{n}(x_i - \overline{x})(y_i - \overline{y})}{\sqrt{\sum_{i=1}^{n}(x_i - \overline{x})^2 \sum_{i=1}^{n}(y_i - \overline{y})^2}} \tag{2}$$

$$\text{MAE} = \frac{1}{n}\sum_{i=1}^{n}|x_i - y_i| \tag{3}$$

$$\text{MRE} = \frac{1}{n}\sum_{i=1}^{n}(\text{MAE}/y_i) \tag{4}$$

$$\text{RMSE} = \sqrt{\frac{1}{n}\sum_{i=0}^{n}(x_i - y_i)^2} \tag{5}$$

In the formula, $x_i$ is the test value of the sample, $y_i$ is the reference value of the sample, $\overline{x}$ and $\overline{y}$ are the multi-year averages of $x_i$ and $y_i$, respectively, and $n$ is the number of samples.

### 3.3. Machine Learning Algorithm

(i)    Random forest

Random forest [28,29] (RF) is a forest composed of decision trees in a random way. There is no correlation between each decision tree of random forest. After obtaining the forest, when a new input sample enters, each decision tree in the forest is judged and classified separately. RF uses the CART decision tree as a weak learner. When generating each tree, the selected features of each tree are only a few randomly selected features, which ensures the randomness of the features. Because of randomness, it plays a very important role in reducing the variance in the model, so random forest generally does not need additional pruning, that is, better generalization ability and anti-overfitting ability can be obtained. On the basis of the decision tree model, multiple training subsets $\{x_1, x_2, \ldots, x_n\}$

are obtained by random sampling of the whole training sample set $\{I_1, I_2, \ldots, I_n\}$, and a decision tree model is trained by using the training subset obtained by each sampling. By constructing a new training data set, multiple decision trees can be trained in parallel, and the integration of multiple decision tree models can be realized. For the test set samples $X$, the multiple prediction values $\{f_1(X), f_2(X), \ldots, f_n(X)\}$ obtained by many decision trees are taken as the final simulation results.

(ii)    BP neural network

A back propagation neural network (BP) is a kind of multi-layer feedforward neural network [30]. Its main characteristic is that the signal propagates forward and the error propagates back. It simulates the structure of the neural network of the human brain and is a simplified biological model. Each layer of neural network is made up of neurons, and each neuron alone is equivalent to a perceptual device. The input layer is single-layer structure, the output layer is also single-layer structure, and the hidden layer can have multiple layers or a single-layer structure. The neurons between the input layer, the hidden layer and the output layer are all connected to each other and are fully connected. In general, the BP neural network structure is that the input layer is stimulated and transmitted to the hidden layer, and the hidden layer will transmit the stimulus to the output layer according to the weights of the neurons and according to the rules. If the accuracy requirements are not met, the output layer will return to the weight of the relationship between the adjusted neurons until the predicted output value of the training meets the target accuracy requirements.

(iii)   Extreme learning machine

The extreme learning machine (ELM) is used to train the single hidden layer feedforward neural network [31]. The hidden layer bias, input layer weight, weight and threshold of the limit learning machine are randomly generated. In the whole training process, it is no longer adjusted. The loss function composed of the training error term and the regular term of the output layer weight norm is minimized to obtain the output layer weight, which is obtained according to Moore–Penrose (MP) generalized inverse matrix theory.

*3.4. Water Balance Analysis*

For closed watersheds, the water balance formula can be written as follows:

$$\Delta W = P - R - ET + W_{\text{Water transfer}} + W_{\text{Error}} \tag{6}$$

In this study, $\Delta W$ is the change value of water storage in the basin, which can be expressed by the value of terrestrial water storage change (TWSC). $P$ is precipitation and $R$ is the total runoff of the basin, including surface runoff (**Rs**) and underground runoff (**Rg**). $ET$ is the actual evapotranspiration of the basin. $W_{\text{Water transfer}}$ is water transfer across watersheds and $W_{\text{Error}}$ is the absolute error of water balance. Therefore, the above formula can be written as follows:

$$TWSC = P - Rs - Rg - ET + W_{\text{Water transfer}} + W_{\text{Error}} \tag{7}$$

## 4. Results

*4.1. Evaluation and Extension of Evaporative Data*

4.1.1. Evaluation of Evaporated Data

Figure 2 is the scatter diagram between ET calculated by TEDAC, GLEAM and GLDAS and the observed data from nine flux stations. Table 2 shows their correlation coefficients and root mean square errors. The $r$ values between ET calculated by TEDAC and ET from flux stations are 0.69~0.96, and the $r$ values between GLDAS data and flux stations data are 0.73~0.97. Except for the Yucheng station and Daxing station, the above $r$ values in other stations are more than 0.8. The results show that the evapotranspirative data simulated by TEDAC and GLDAS are in good agreement with those from most flux stations. The $r$ values between ET calculated by TEDAC and ET data from three grassland stations

(Arou station, Yakou station and Haibei station) are all above 0.9, which indicates that the application effect of TEDAC products in semi-arid region of grassland is better than other kinds of regions. Only one-third of the farmland stations have *r* values above 0.9, and Luancheng station has an obvious underestimation., which indicates that the simulation effect of the TEDAC data set is general in the area of farmland. The *r* values between GLEAM data and flux stations data are 0.76~0.96, and the *r* values of all stations are more than 0.75. The *r* values of all grassland stations are above 0.9, indicating that the application effect of GLEAM products in the grassland semi-arid area is also very good, and the correlation between GLEAM data and data from Daxing station and Yucheng station has been improved compared with TEDAC data sets and GLDAS data sets.

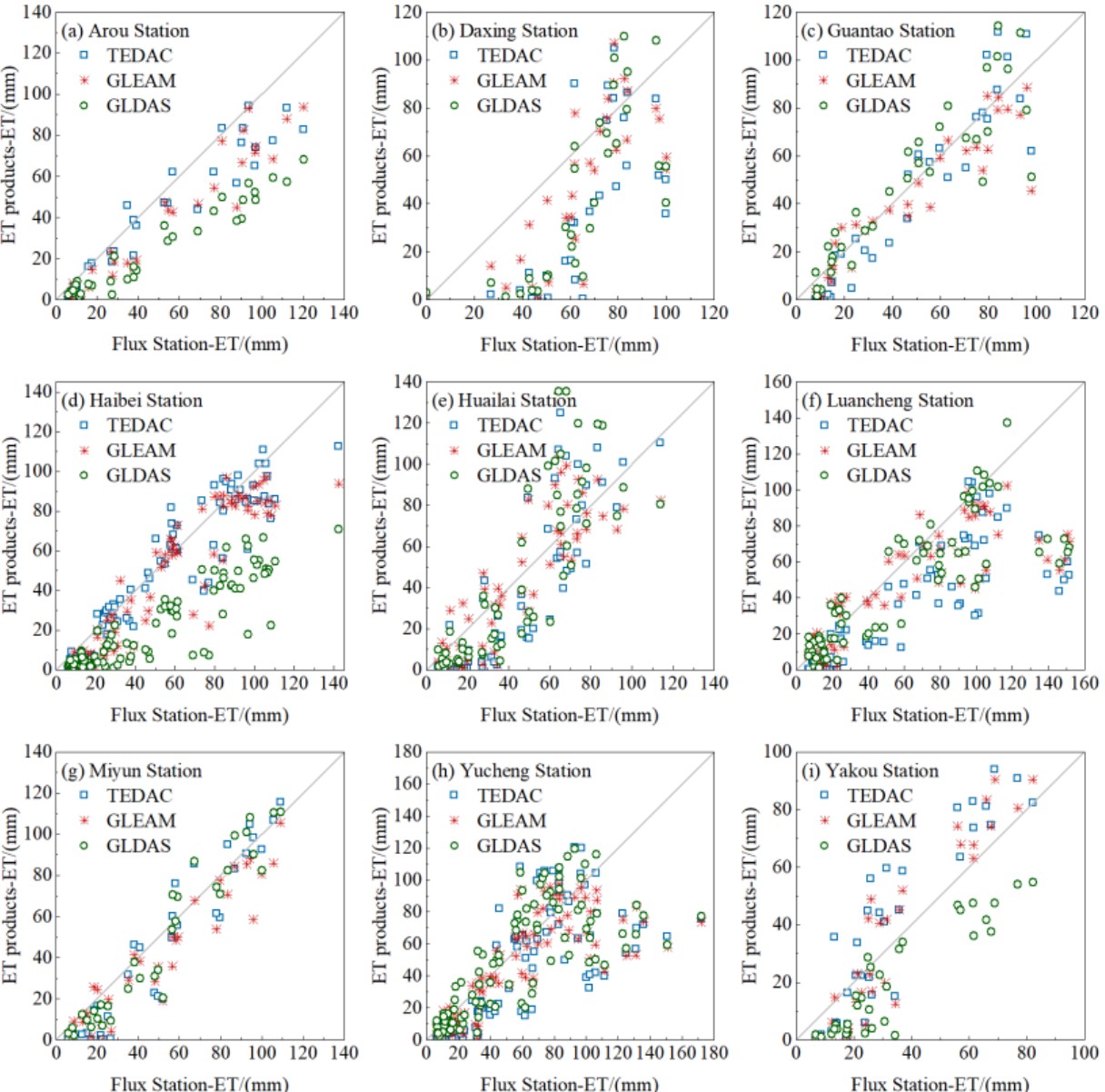

**Figure 2.** Correlation between the values of three evapotranspiration products and evapotranspiration observed at flux stations.

**Table 2.** Correlation coefficient and RMSE between the values of evapotranspiration products and the evapotranspiration at the flux site.

| Site | *r* | | | RMSE | | |
|---|---|---|---|---|---|---|
| | TEDAC | GLEAM | GLDAS | TEDAC | GLEAM | GLDAS |
| Arou station | 0.96 | 0.96 | 0.97 | 15.03 | 18.08 | 30.28 |
| Daxing station | 0.69 | 0.76 | 0.74 | 36.57 | 26.58 | 32.40 |
| Guantao station | 0.95 | 0.93 | 0.90 | 12.83 | 12.74 | 14.51 |
| Haibei station | 0.95 | 0.94 | 0.89 | 12.41 | 15.44 | 33.80 |
| Huailai station | 0.84 | 0.84 | 0.82 | 21.69 | 17.14 | 23.31 |
| Luancheng station | 0.82 | 0.83 | 0.82 | 35.27 | 28.69 | 28.09 |
| Miyun station | 0.95 | 0.94 | 0.96 | 14.43 | 14.64 | 11.43 |
| Yucheng station | 0.73 | 0.78 | 0.73 | 31.50 | 26.85 | 28.11 |
| Yakou station | 0.90 | 0.94 | 0.92 | 15.38 | 12.95 | 16.40 |

We know that the error of evapotranspiration data simulated by GLDAS is large by the RMSE, which is lower than 20 mm in only three stations (Guantao station, Miyun station and Yakou station). The errors of TEDAC products and GLEAM products are large in Daxing station, Luancheng station and Miyun station, but the errors of GLAEM products are maintained below 20 mm, while those of TEDAC products exceed 30 mm. Although TEDAC products are superior to GLAEM products in some stations, there is no significant difference between the two products. Therefore, on the whole, the application effect of GLEAM evapotranspirative products in HRB is the best and we can use this data set to reflect the evolution of ET.

4.1.2. Extension of Evapotranspiration Data

The effects of meteorological and vegetation driving factors, such as precipitation, relative humidity, air temperature, sunshine time, wind speed and NDVI, on the ET are considered synthetically. The spatial scale is each third grade area in the basin, while the time scale is monthly date, and the sequence length of meteorological data is 1956–2018.

We use the correlation coefficient to analyze the correlation between the driving elements of the eight regions and between the driving elements and ET. The results are shown in Figure 3. In the Haihe mountainous area, the correlation coefficient between sunshine and ET is low, and so is wind speed. Precipitation, humidity, NDVI, sunshine time and temperature are significantly and positively correlated with ET ($p < 0.01$). The three driving elements, precipitation, NDVI and air temperature is the main influencing factor of ET. The situation of Haihe plain is similar to that of the Haihe mountainous area. For the area above Lanzhou, there is a significant positive correlation between temperature and ET and also NDVI and ET, just as the *r* value is above 0.8. The correlation coefficient between humidity and ET is also higher than that in the Haihe River Basin, which reaches 0.71. In the Lanzhou-Toudaoguai Interval, humidity and wind speed have a weak positive correlation with ET, while precipitation, temperature and NDVI have a significant positive correlation with ET, which are the main factors affecting ET. For the Lower Yellow River, wind speed and sunshine have little influence on ET; therefore, precipitation, temperature, NDVI and humidity are the main influencing factors. For the Huaihe River Basin and Shandong Peninsula, the correlation coefficient between temperature and ET is the highest, which is more than 0.9. Precipitation, sunshine, NDVI and humidity are also significantly and positively correlated with ET. These five elements will have a great impact on the actual evaporation. On the whole, precipitation, humidity, temperature, NDVI and sunshine have greater influence on ET, while the influence of wind speed can be ignored.

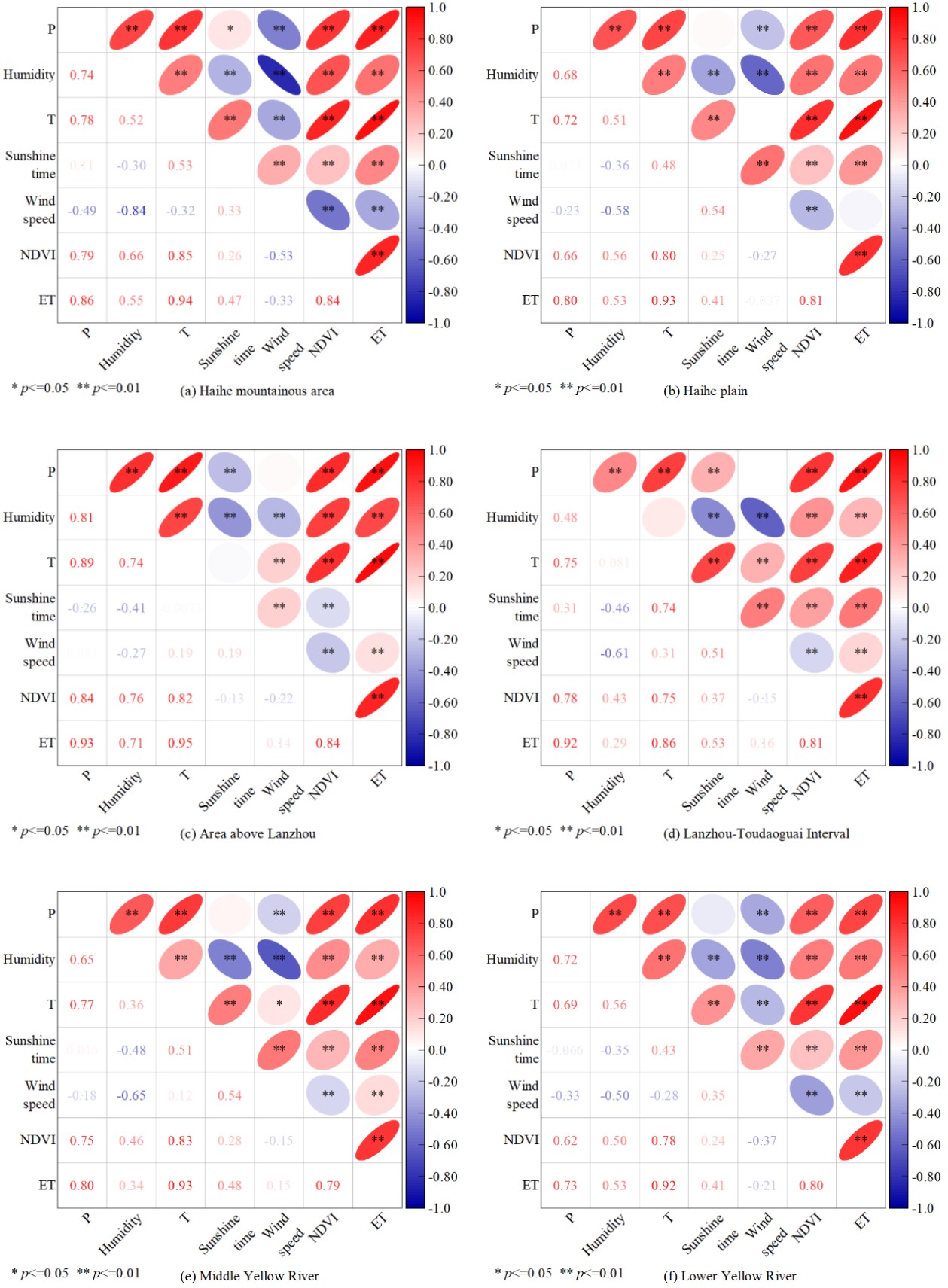

**Figure 3.** *Cont.*

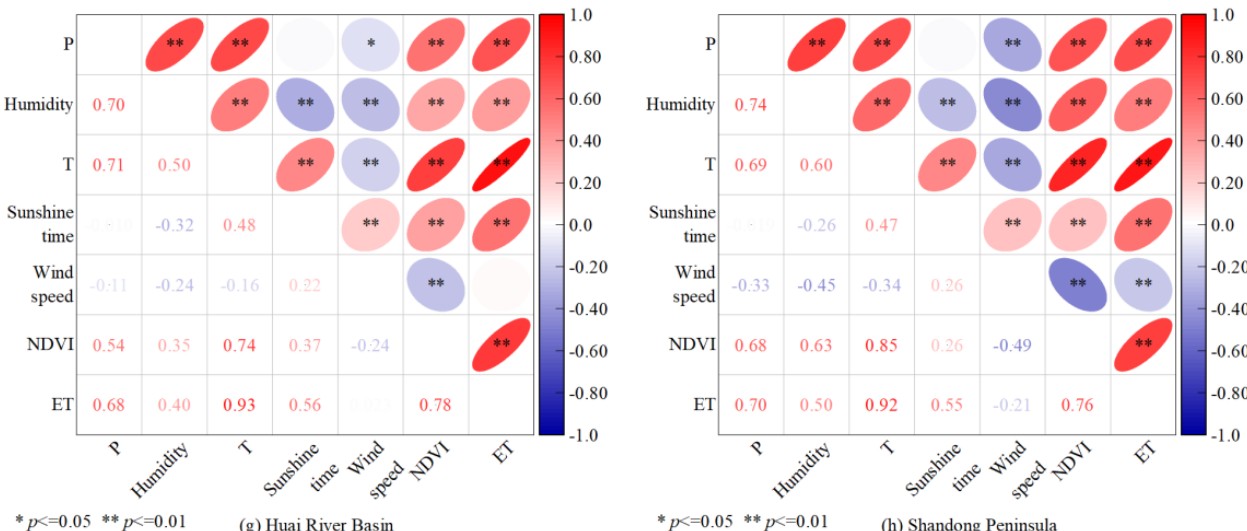

**Figure 3.** Pairwise correlation coefficients between driving factors and actual evapotranspiration in the HRB.

Combined with the results of the correlation analysis, we decided to use precipitation, relative humidity, air temperature, sunshine time and NDVI as driving factors. Although the NDVI data before 1982 cannot be obtained, according to the research, it is found that the NDVI of HRB has increased obviously only after 2000, due to the conversion of cropland to forest and grassland and conservation projects of soil and water resource, and before 2000, the change in NDVI is relatively small. Therefore, the monthly mean value of NDVI in eight regions from 1982 to 2000 represents NDVI from 1956 to 1979, and then the driving factors of machine learning model are analyzed.

The construction of the actual evaporative calculation model is based on the analysis results of the actual evaporative driving elements, each element data set is taken as the input of the model, and the number of training set and test set are selected. The ET of the eight regions is used as the output of the model. Three different machine learning algorithms are used to train and test the model, respectively, and then we can obtain the simulation results. Each driver element set cover monthly scale data from 1982 to 2018, 80% of which are selected by the training set and 20% are the test set.

The three machine learning methods work well on the training set and the test set, because the fitting coefficients $r^2$ of all regions have reached about 0.9. The correlation of the area above Lanzhou is the best, according to the $r^2$ (0.98). The simulation results in the Lower Yellow River and Shandong Peninsula are poor, but the coefficients are still close to 0.9. The $r^2$, RMSE and MRE of the test set are shown in Table 3 and Figure 4. In the test set, the prediction accuracy of various machine learning algorithms also remains at a high level. The root mean square error is 4.0~10.5, and the average relative deviation is about 28.9%. Combined with the results of $r^2$, RMSE and MRE, the simulation effect of the random forest algorithm in HRB is the best because the simulation effects of the BP neural network and extreme learning machine are better, which are only embodied in some individual areas. So, we choose the simulation results of random forest to invert the actual evapotranspiration in HRB from 1956 to 1979.

**Table 3.** Comparison of simulation results of machine learning test set.

| Research Area | RF | | | BP | | | ELM | | |
|---|---|---|---|---|---|---|---|---|---|
| | $r^2$ | RMSE | MRE | $r^2$ | RMSE | MRE | $r^2$ | RMSE | MRE |
| Haihe mountainous area | 0.94 | 9.10 | 28.9% | 0.94 | 6.32 | 38.8% | 0.95 | 6.01 | 46.6% |
| Haihe plain | 0.94 | 7.14 | 13.6% | 0.91 | 7.98 | 33.8% | 0.92 | 7.95 | 48.1% |
| Area above Lanzhou | 0.98 | 7.85 | 21.7% | 0.98 | 4.08 | 21.8% | 0.98 | 4.00 | 20.3% |
| Lantou Interval | 0.92 | 6.46 | 18.0% | 0.91 | 4.59 | 31.5% | 0.91 | 4.72 | 23.1% |
| Middle Yellow River | 0.93 | 4.02 | 27.1% | 0.92 | 6.98 | 23.2% | 0.90 | 7.51 | 27.8% |
| Lower Yellow River | 0.92 | 4.55 | 19.7% | 0.88 | 9.95 | 27.4% | 0.88 | 10.13 | 40.2% |
| Huai River Basin | 0.94 | 6.29 | 23.5% | 0.93 | 7.60 | 18.0% | 0.92 | 8.63 | 26.9% |
| Shandong Peninsula | 0.92 | 6.19 | 21.5% | 0.89 | 10.50 | 56.3% | 0.91 | 9.61 | 37.9% |

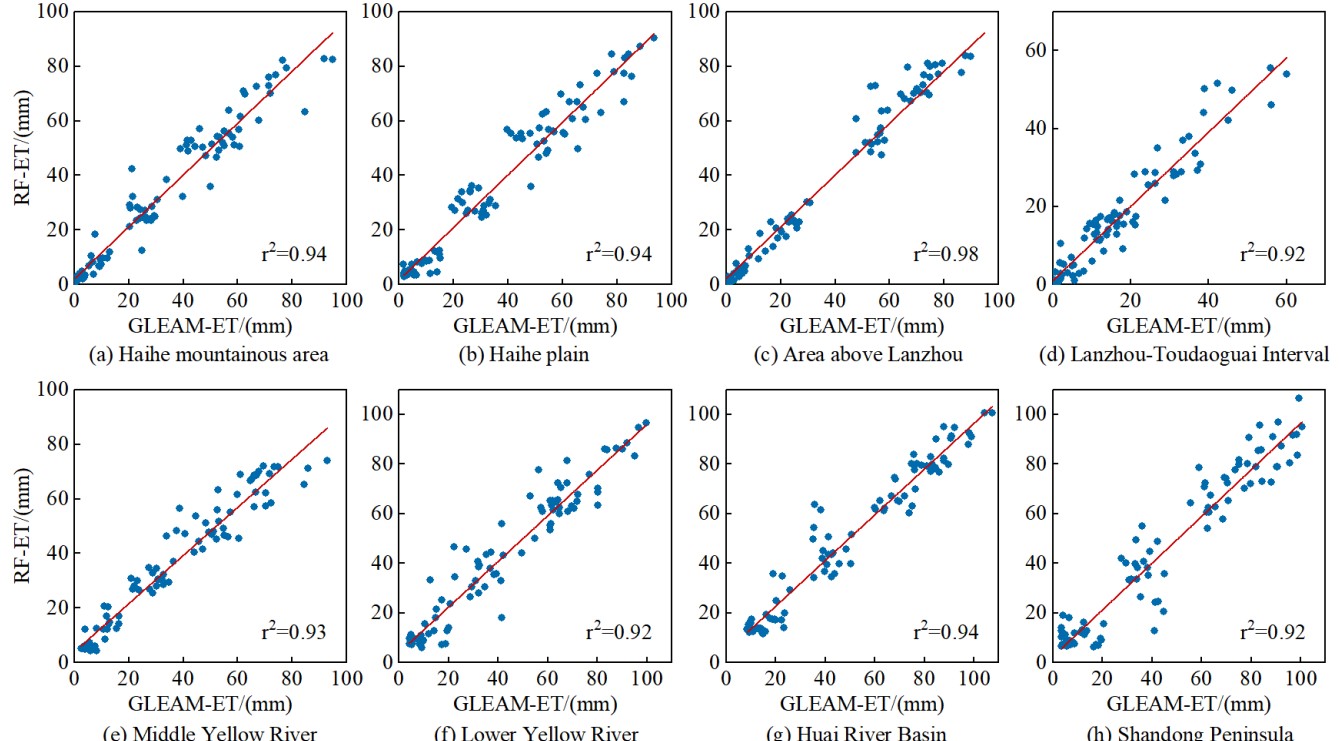

**Figure 4.** Random forest simulation results.

*4.2. Regional Water Cycle Change*

4.2.1. Evolution of Hydrological Elements

Figure 5 shows the evolution of hydrological elements in HRB and its subregions from 1956 to 2018. The total precipitation in HRB from 1956 to 2018 is between 576.3 and 1080.6 billion m³, and the total evaporation is 554.3~736 billion m³, and the total runoff is 128.1~356.8 billion m³. The absolute error of water balance is between −148.1 and 29.7 billion m³. The precipitation fluctuates greatly, and the precipitation is less than evaporation in individual years in the Haihe Plain and the Lower Yellow River, which is the main agricultural area in North China. Affected by artificial agricultural irrigation, it will lead to the increase in evapotranspiration in the basin.

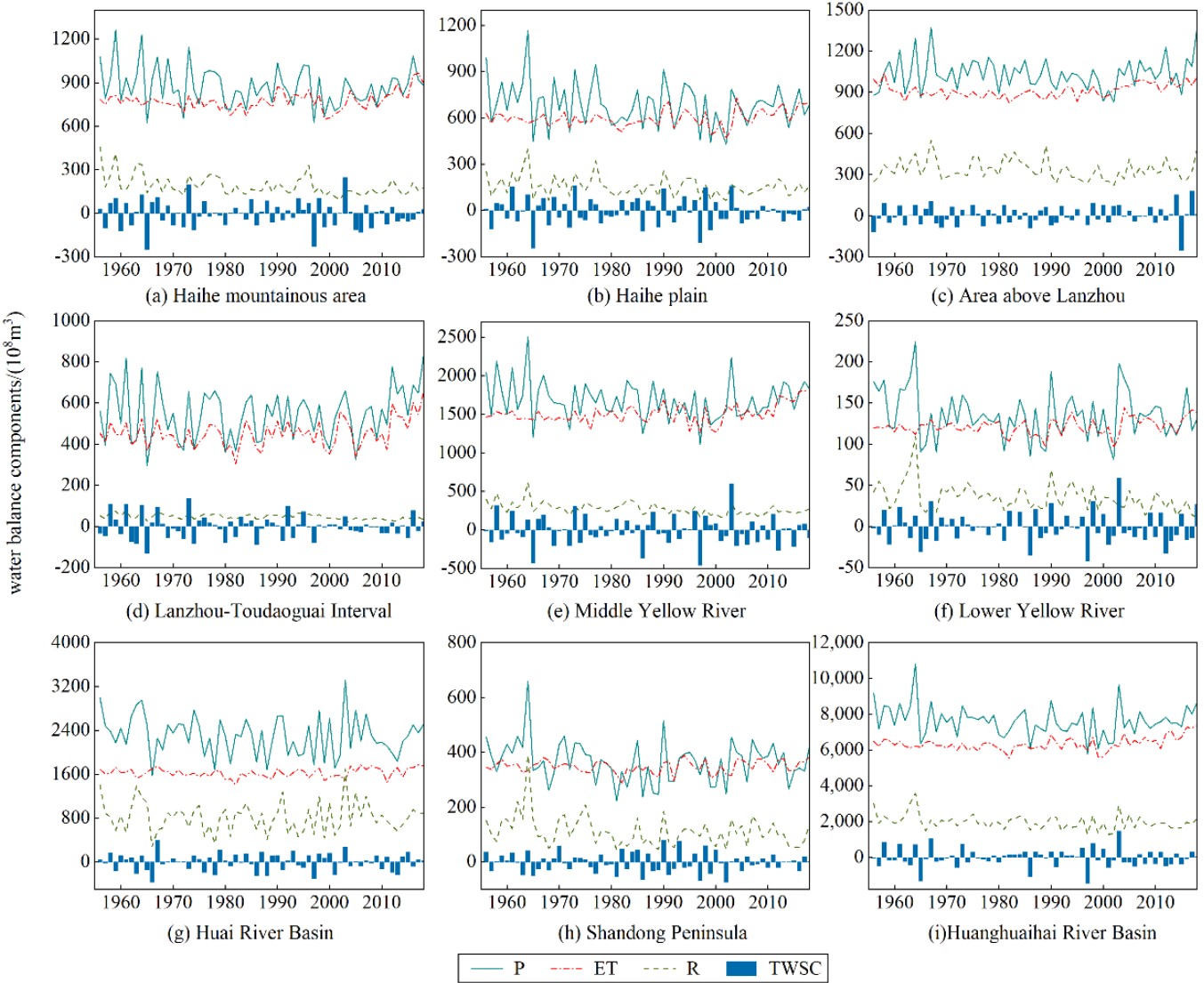

**Figure 5.** Evolution of water balance components in the HRB from 1956 to 2018.

Figure 6 shows the temporal variation trend of water cycle elements in eight sub-regions of HRB, in which the precipitation in the Lanzhou-Toudaoguai Interval shows no significant upward trend, while the rest of the area shows a downward trend. The evaporation has a significant upward trend in the whole Yellow River Basin and Haihe Plain, but no significant change in other areas. The amount of surface water resources in Haihe mountainous area and plain, the middle and lower reaches of the Yellow River and Shandong Peninsula all show a significant downward trend, which show a significant upward trend in the area above Lanzhou and Haihe mountainous areas, and the change in TWSC is relatively smooth, embodied by the insignificant upward or downward trend.

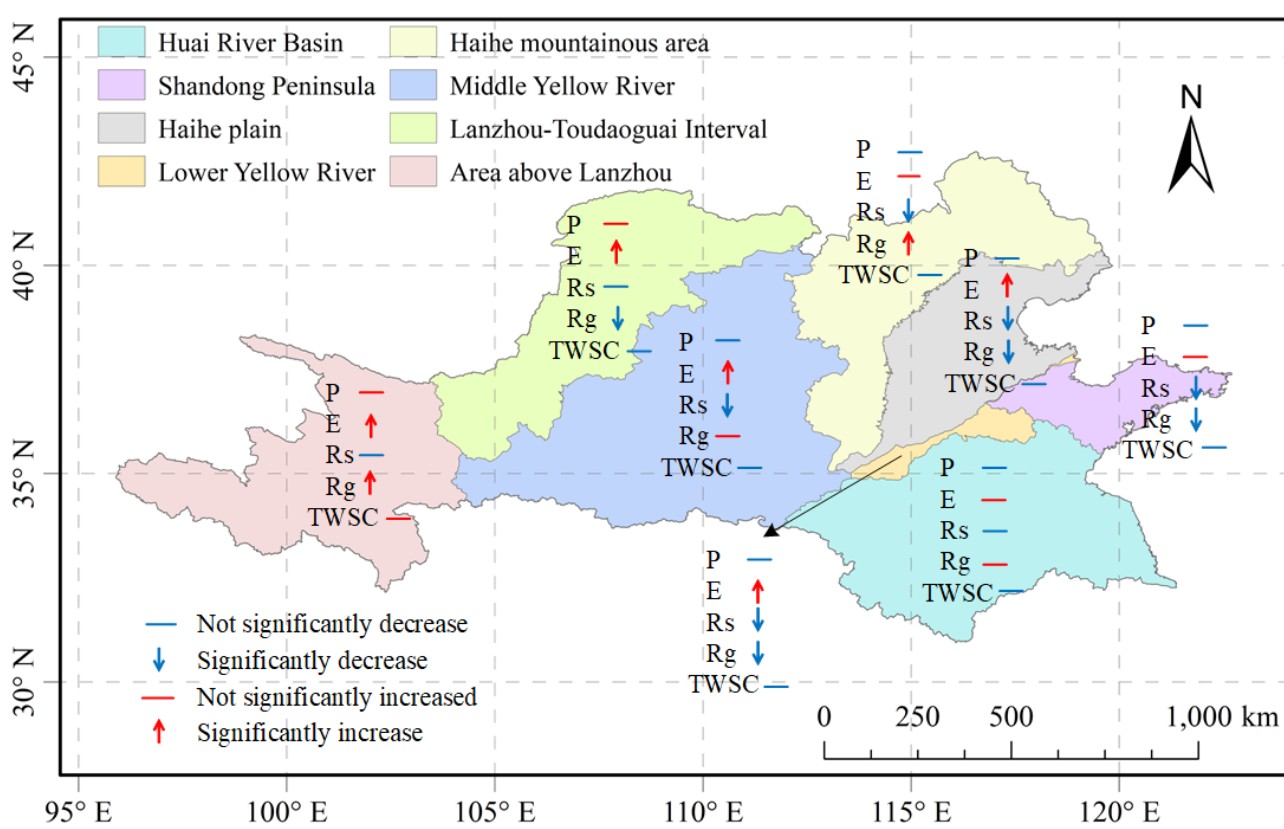

**Figure 6.** Temporal variation trend of water cycle elements in the HRB from 1956 to 2018.

### 4.2.2. Variation in Water Balance

According to the study of the change in hydrological elements, the precipitation has obviously obviously since 1980, and the precipitation in each district has evidently recovered in 2003. Therefore, taking 1980 and 2003 as the two time nodes, the water balance analysis is divided into three stages. Table 4 shows the water balance calculations of eight study areas in HRB. From the table, we can observe that in addition to the Lantou Interval, the amount of income is less than the amount of water expenditure in the other areas. The main expenditure item of water quantity in each study area is evaporation, and it accounts for the largest proportion in the Lantou Interval, which is about 89.2~93.4%, this situation may be caused by drought and shortage of water resources in the area. Meanwhile, evaporation accounts for the least proportion in the Huaihe River Basin, where ET accounts for only 65.2~67.6% of the total water expenditure. The Huaihe River Basin is rich in precipitation and has a high population density and large amount of water demand for social production, life and agriculture, resulting in a small proportion of evapotranspiration in contrast to the total expenditure. The variation value of TWSC appears to be positive in the Lanzhou area. Combined with other scholars, it is found that the reason may be the degradation of permafrost caused by the increase in temperature, which leads to the increase in water storage capacity of soil aquifer, and then it changes the hydrological cycle process in the study area. In short, permafrost degradation may affect runoff by increasing groundwater reserves. When the thickness of the active layer in permafrost area increases, more surface water will infiltrate into underground aquifer through infiltration, which will lead to the increase in TWS in the area above Lanzhou.

The water balance analysis of HRB includes the calculations of income water quantity, expenditure water quantity, basin water storage capacity and their changes, in which the main sources of income water quantity are precipitation and external water transfer, and the expenditure items include actual evaporation, surface runoff and underground runoff, while the storage variables are reflected by the change value of land water reserves. Table 5

and Figure 7 show the water balance of HRB from 1956 to 2018. The total income water quantity of the HRB from 1956 to 1979 was 793.38 billion m$^3$. The total amount of water expenditure is 841.71 billion m$^3$. The absolute error of water balance is $-35.61$ billion m$^3$. From 1956 to 1979, the relative error was $-4.5\%$. The total income of the HRB from 1980 to 2002 was 720.09 billion m$^3$. The total amount of water expenditure is 799.29 billion m$^3$. The absolute error of water balance shows little change. From 1980 to 2002, the relative error of water balance is $-4.91\%$, and the error has increased slightly. The total income of the HRB from 2003 to 2018 was 785.66 billion m$^3$. The total amount of water expenditure is 863.21 billion m$^3$. The absolute error of water balance is 38.53 billion m$^3$. In all years, the relative errors of water balance in 2004, 2006, 2009, 2013 and 2017 are larger, which in the rest of the years are less than 10%. The relative errors of water balance in the three research stages are all within $\pm5\%$, which indicates that the calculation of each element in the water cycle of HRB has reached a relatively good accuracy and the calculation results can reflect the water balance relationship of basin correctly.

**Table 4.** Statistical unit of water balance entry and exit and storage variables in HRB (100 million m$^3$).

| Zoning | Years | Income Water Quantity | Expenditure Water Quantity | | | | Water Storage Change Value |
|---|---|---|---|---|---|---|---|
| | | $p$ | ET | Rs | Rg | Total | TWSC |
| Haihe mountainous area | 1956–1979 | 924.2 | 764.8 | 187.4 | 45.1 | 997.3 | −21.8 |
| | 1980–2002 | 827.5 | 754.3 | 119.5 | 48.4 | 922.2 | −118.7 |
| | 2003–2018 | 869.4 | 818.9 | 96.2 | 61.5 | 976.6 | −208.7 |
| Haihe plain | 1956–1979 | 737.2 | 592.4 | 68.5 | 116.5 | 777.4 | −97.5 |
| | 1980–2002 | 639.0 | 576.4 | 42.8 | 94.0 | 713.2 | −88.7 |
| | 2003–2018 | 683.0 | 640.5 | 43.8 | 98.9 | 783.3 | −164.9 |
| Area above Lanzhou | 1956–1979 | 1050.3 | 904.7 | 346.4 | 1.3 | 1252.5 | 3.9 |
| | 1980–2002 | 991.7 | 885.5 | 322.6 | 3.1 | 1211.2 | 42.7 |
| | 2003–2018 | 1075.2 | 963.1 | 333.8 | 1.8 | 1298.8 | 181.5 |
| Lantou Interval | 1956–1979 | 553.2 | 443.9 | 21.2 | 32.6 | 497.6 | 57.0 |
| | 1980–2002 | 513.0 | 434.4 | 19.2 | 29.3 | 482.8 | −80.0 |
| | 2003–2018 | 589.7 | 491.2 | 17.6 | 16.8 | 525.6 | −63.8 |
| Middle Yellow River | 1956–1979 | 1748.5 | 1468.7 | 254.1 | 63.8 | 1786.7 | −76.3 |
| | 1980–2002 | 1590.5 | 1479.1 | 196.6 | 63.5 | 1739.2 | −90.9 |
| | 2003–2018 | 1721.0 | 1618.9 | 171.7 | 69.3 | 1859.9 | −162.1 |
| Lower Yellow River | 1956–1979 | 142.5 | 120.8 | 26.3 | 16.3 | 163.4 | −14.3 |
| | 1980–2002 | 125.5 | 117.2 | 20.6 | 14.3 | 152.0 | −14.7 |
| | 2003–2018 | 140.2 | 128.8 | 11.8 | 13.7 | 154.2 | −12.4 |
| Huaihe River Basin | 1956–1979 | 2384.8 | 1633.0 | 610.9 | 207.2 | 2451.1 | 18.4 |
| | 1980–2002 | 2184.9 | 1584.2 | 556.7 | 203.1 | 2343.9 | −35.1 |
| | 2003–2018 | 2345.0 | 1667.1 | 663.2 | 230.3 | 2570.6 | 66.9 |
| Shandong Peninsula | 1956–1979 | 393.0 | 351.1 | 103.9 | 36.1 | 491.2 | 3.6 |
| | 1980–2002 | 328.9 | 338.9 | 57.3 | 32.1 | 428.4 | −52.8 |
| | 2003–2018 | 377.4 | 358.2 | 76.6 | 28.4 | 463.2 | −26.8 |

**Table 5.** Analysis of water balance in HRB from 1956 to 2018.

| Year | Total Income Water Quantity ($10^8$ m$^3$) | Total Expenditure Water Quantity ($10^8$ m$^3$) | Water Storage Change Value ($10^8$ m$^3$) | Absolute Error ($10^8$ m$^3$) | Water Balance Relative Error |
|---|---|---|---|---|---|
| 1956–1979 | 7933.8 | 8417.1 | −127.1 | −356.1 | −4.49% |
| 1980–2002 | 7200.9 | 7992.9 | −438.3 | −353.7 | −4.91% |
| 2003–2018 | 7856.6 | 8632.1 | −390.3 | −385.3 | −4.94% |

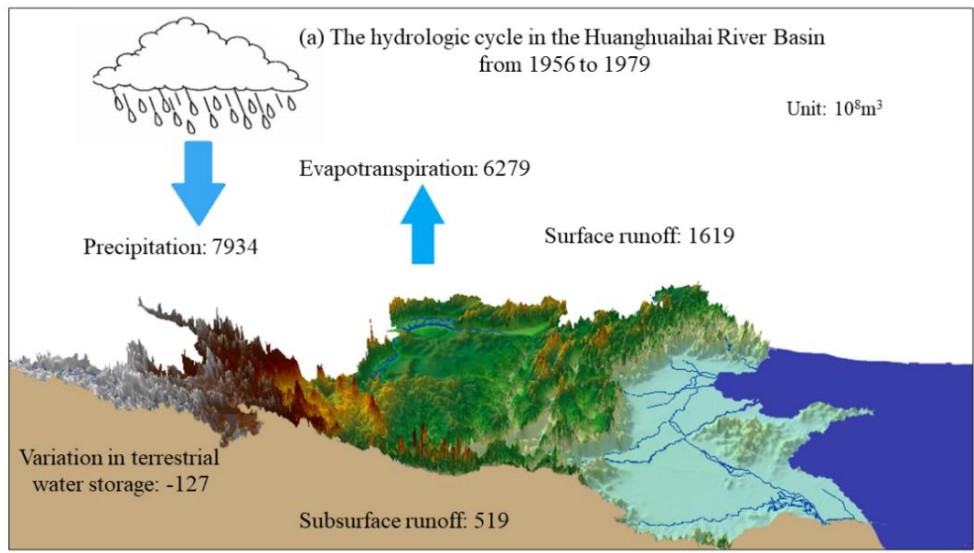

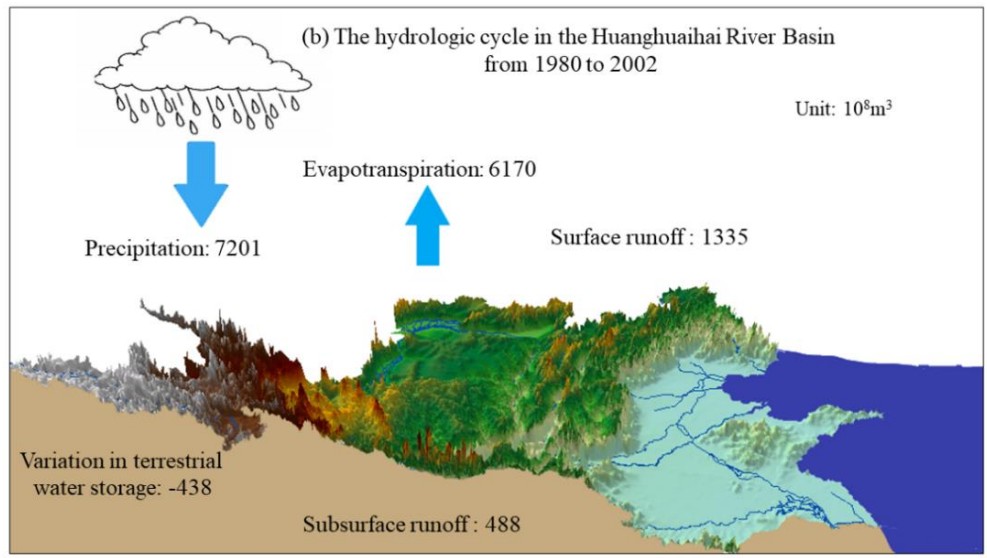

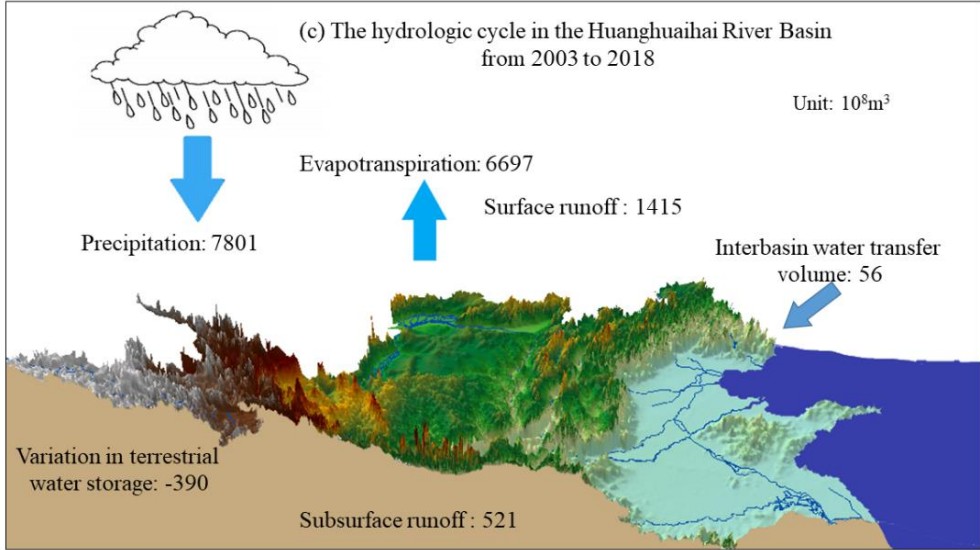

**Figure 7.** Diagram of water cycle in the HRB.

## 5. Discussion

The study of the cause and space pattern of ET has always been a hot issue in the study of the hydrological cycle. Because of the few observation stations and short observation time in the HRB, the research based on the measured data has not yet reached a unified conclusion on the actual evaporation in the study area. The Mann–Kendall test is not suited for data with periodicities (i.e., seasonal effects). Other robust season-trend fit models, such as least-squares spectral analysis, could also be useful [32]. At present, the average annual evaporation of different products in China is quite different [33,34]. As far as the changing trend is concerned, the results of MOD16 and model simulation show that the ET in China is mainly on the rise [35–37]. Compared with other products, GLEAM products have better accuracy and spatial resolution in the Huaihe River Basin [38], Yellow River source region [39] and even the whole of China [40,41].

When simulating evaporation by machine learning, this paper focuses on meteorological and vegetation driving factors, such as precipitation, temperature and NDVI, but the terrain of HRB is complex and the climate change is diverse. In recent years, the change in underlying surface conditions caused by human activities, such as cross-basin water transfer and afforestation, will also affect the calculation of actual evaporation [42,43]. The correlation between meteorological factors and ET in the Haihe River Basin and the middle reaches of the Yellow River is poor or negative, which may be the influence of urban development and industrialization, and the phenomenon of "evaporation paradox" appears [44,45]. In addition, some studies show that the accuracy of the integration results obtained by using a variety of algorithms is higher than the estimated value of a single ET model [46,47]. Therefore, combined with a number of control factors, the fusion of the actual evaporation simulation method is one of the research priorities in the future.

For the calculation results of the water balance in HRB from 1956 to 2018, it is found that the amount of water earned in the study period is less than the amount of water spent, and cannot reach a complete balance, and the amount of water in revenue is less than the amount of water spent. According to the water resources bulletin of the basin, the annual average annual precipitation of HRB from 2003 to 2018 is 825.2 billion $m^3$. The relative error of water balance calculated according to this value is only 0.8%, which significantly improves the accuracy of the water balance calculation. Due to the lack of water transfer data, this paper only considers the impact of cross-basin water transfer after 2008, which will also lead to low income and water volume. In addition, when calculating the change value of water storage, the changes in reservoir operation and deeper soil water are not taken into account. Only 0~2 m soil water is considered in the calculation of water reserves [48], while there are many reservoirs in HRB, the degree of groundwater development and utilization is high, and the groundwater level evidently drops [49], which is also an important reason for the imbalance of water quantity.

## 6. Conclusions

In order to study the change in the water cycle in HRB, the evolution law and influencing factors of ET were analyzed. Combining GLEAM products with vegetation and meteorological data, the ET from 1956 to 1980 in HRB is simulated by the machine learning algorithm. Based on the calculation results of each component of the water balance formula, the following conclusions are obtained:

(1) The accuracy verification results of ET data from flux stations show that the simulation accuracy of GLEAM products in HRB is good, and the annual average ET spatial distribution tends to increase from northwest to southeast as a whole.

(2) Precipitation, temperature, sunshine, humidity and NDVI are used as the influencing factors of ET. Three machine learning algorithms, BP, RF and ELM, are used to construct the ET calculation model. The *r* values of the three methods are all above 0.9 and the MRE values of RF are all less than 30%. The RF calculates the best results.

(3)　The water balance analysis of HRB shows that the income item of the basin is mainly precipitation, and the amount of water spent is mainly ET, accounting for 76% of the total amount of water spent, and TWSC of the basin is in a state of decrease. The relative errors of water balance in HRB from 1956–1979, 1980–2002 and 2003–2018 are less than ±5%, which indicates that the calculation of each element of the water cycle in HRB has reached a certain accuracy and can reflect the relationship of water balance in HRB.

At present, there are many products for the research on ET and the integration of suitable products should be considered to improve the accuracy of the calculation results. Different data sources cause certain uncertainties in the results of water balance studies. How to realize the fusion of data and the elimination of unbalanced quantities in the watershed still needs to be studied.

**Author Contributions:** Conceptualization and methodology, X.Y.; formal analysis, C.L. and G.W.; data curation, J.J. and G.W.; writing—original draft, X.Y.; writing—review and editing, Z.B.; and visualization, Z.B. All authors have read and agreed to the published version of the manuscript.

**Funding:** This research was funded by the National Key Research and Development Program of China (grant numbers 2017YFA0605002, 2017YFA0605004), the National Natural Science Foundation of China (grant numbers 41961124007, 51779145, 41830863), and the "Six top talents" in Jiangsu province (grant number RJFW-031).

**Data Availability Statement:** Data are available based upon request to corresponding author.

**Conflicts of Interest:** The authors declare no conflict of interest.

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
