# Peer review of "Trends and Changes in Hydrologic Cycle in the Huanghuaihai River Basin from 1956 to 2018"

_water, doi:10.3390/w14142148_

Round 1
Reviewer 1 Report
This work explored a trend and change in hydrologic cycle in the Huanghuaihai River Basin. Overall, the work is interesting and informative, and some new findings have been investigated. It told us how the hydrologic cycle changed during the last several decades. Generally, the manuscript was clearly organized. The introduction has a good literature review, the study procedure is appropriate, and the results are reasonable. I recommend acceptation of this manuscript with a minor revision.
Some specific comments are list as flow:
1. Abstract should be written in concise and quantitative. I would suggest listing only some of the most important results to justify the implications and conclusions of the study.
2. I highly recommended to authors, if possible, please modify the figure with good quality images.
3. There are some repetitive statements in the analysis of Figures 2 and 3, which can be simplified.
4. Human activities also have an impact on the trend in water cycle. Specific human activities in this region should be discussed.
5. Some of words/ abbreviations need explanation, e.g. line 434 TWSC.

Author Response
Dear editor and reviewer,
All authors do appreciate anonymous referees for their great efforts on the manuscript reviewing. All comments are very valuable and constructive for us to improve the manuscript quality. We improved the manuscript based on fully consideration of these comments. Responses to these comments are given in the Word file.

Reviewer 2 Report
Reviewer’s Report on the manuscript entitled:
Trends and changes in hydrologic cycle in the Huanghuaihai River Basin from 1956 to 2018
The authors applied machine learning methods to prolong the actual evapotranspiration of the Huanghuaihai River Basin on the time scale and explore water balances calculated from various sources. The manuscript is well-structured and the results are interesting, but there are some grammar issues in the manuscript. Please see below my comments:
Please define all the acronyms the first time they appear in both Abstract and Manuscript and be consistent with their style. Please also add an acronym table at the end of the manuscript listing all acronyms used.
Line 80. Please define an acronym like HRB for HuangHuaiHai River Basin and use the acronym HRB in other places.
Lines 37-41. Grammar issue. Please re-write.
Line 43. Replace “It’s” with “It is”.
Line 44. equations or equation? See line 42.
Lines 54-55. Grammar issue. Please re-write.
Line 94. Please replace “It is of great significance” with “It is crucial”
Line 96. By referring to the Section numbers please describe how the rest of the manuscript is organized.
Line 101. Please fix the reference
Lines 153, 163, 190, etc. Please move the links to the References and write the last time when you successfully accessed the websites and cite them as [x] in the manuscript as you did for other references.
Line 231. Please define NDVI and cite the following articles describing them:
https://doi.org/10.1080/01431161.2019.1688419
Rouse, J.W.; Hass, R.H., Jr.; Schell, J.A.; Deering, D.W. Monitoring vegetation systems in the great plains with ERTS. In Proceedings of the Third Earth Resources Technology Satellite-1 Symposium 1 (A), Texas A&M University, College Station, TX, USA, 1 January 1974; pp. 309–3
Equations (3) and (4). MAE has different styles.
Line 303. Please merge this line to the previous sentence.
Finally, please carefully proofread the manuscript.
Line 540. Could you mention here some numerical comparisons for different methods such as the overall accuracy of each method to justify your statement?
In the discussion part please discuss the limitation of MK test:
The Mann-Kendall test is not suited for data with periodicities (i.e., seasonal effects). Other robust season-trend fit models such as least-squares spectral analysis could also be useful. You may mention the following article below describing the new software being used in hydrology, remote sensing, and others that considers jumps, seasonality, and irregularities in samplings:
https://doi.org/10.1007/s10291-021-01118-x
In Conclusions, please mention the limitations of the study and provide recommendations and future direction.
Thank you for your contribution
Regards,
Author Response

(The authors gave the same response as above.)
